# Mouthguard-Type Wearable Sensor for Monitoring Salivary Turbidity to Assess Oral Hygiene

**DOI:** 10.3390/s24051436

**Published:** 2024-02-23

**Authors:** Kenta Ichikawa, Kenta Iitani, Gentaro Kawase, Koji Toma, Takahiro Arakawa, Dzung Viet Dao, Kohji Mitsubayashi

**Affiliations:** 1Department of Biomedical Devices and Instrumentation, Institute of Biomaterials and Bioengineering, Tokyo Medical and Dental University (TMDU), Tokyo 101-0062, Japan; ichikawa.kenta@tmd.ac.jp (K.I.); i.bdi@tmd.ac.jp (K.I.); k-toma@shibaura-it.ac.jp (K.T.); arakawath@stf.teu.ac.jp (T.A.); 2Graduate School of Medical and Dental Sciences, Tokyo Medical and Dental University (TMDU), Tokyo 113-8510, Japan; 3Department of Electronic Engineering, College of Engineering, Shibaura Institute of Technology, Tokyo 135-8548, Japan; 4Department of Electric and Electronic Engineering, Tokyo University of Technology, Tokyo 192-0982, Japan; 5School of Engineering and Built Environment, Griffith University, Southport 4222, Australia; d.dao@griffith.edu.au; 6Queensland Micro- and Nanotechnology Centre, Griffith University, Nathan 4111, Australia

**Keywords:** wearable sensor, mouthguard, oral hygiene, salivary turbidity, optical sensing

## Abstract

Salivary turbidity is a promising indicator for evaluating oral hygiene. This study proposed a wearable mouthguard-type sensor for continuous and unconstrained measurement of salivary turbidity. The sensor evaluated turbidity by measuring the light transmittance of saliva with an LED and a phototransistor sealed inside a double-layered mouthguard. The sensor was also embedded with a Bluetooth wireless module, enabling the wireless measurement of turbidity. The mouthguard materials (polyethylene terephthalate-glycol and ethylene-vinyl acetate) and the wavelength of the LED (405 nm) were experimentally determined to achieve high sensitivity in salivary turbidity measurement. The turbidity quantification characteristic of the proposed sensor was evaluated using a turbidity standard solution, and the sensor was capable of turbidity quantification over a wide dynamic range of 1–4000 FTU (formazine turbidity unit), including reported salivary turbidity (400–800 FTU). In vitro turbidity measurement using a saliva sample showed 553 FTU, which is equivalent to the same sample measured with a spectrophotometer (576 FTU). Moreover, in vivo experiments also showed results equivalent to that measured with a spectrophotometer, and wireless measurement of salivary turbidity was realized using the mouthguard-type sensor. Based on these results, the proposed mouthguard-type sensor has promising potential for the unconstrained continuous evaluation of oral hygiene.

## 1. Introduction

Evaluation of oral hygiene is instrumental in diagnosing the health status of individuals [1,2]. Poor oral hygiene not only causes oral diseases such as dental caries [3,4] and periodontal disease [5], but also carries the risk of developing various systemic diseases [6]. For example, periodontitis has been reported to be correlated with diabetes and cardiovascular disease, and the mechanism by which periodontitis develops into such a serious systemic disease has also been reported [7,8]. Therefore, continuous evaluation of oral hygiene is expected to contribute to risk assessment, early diagnosis, and preventive medicine for these systemic diseases.

As an evaluation index of oral hygiene, quantification of oral bacteria is widely used [9]. A typical method for quantification of oral bacteria is to culture sampled dental plaque [10] or tongue plaque [11]. However, the culturing process requires time and appropriate equipment, and is not suitable for real-time evaluation of oral hygiene. As a simpler sample for oral hygiene evaluation, mouth-rinsed water has been utilized [12,13]. Takeuchi et al. reported that the turbidity of mouth-rinsed water is a potential indicator of several oral health conditions, including oral hygiene [14]. On the other hand, sampling of mouth-rinsed water requires the external water supply and rinsing process, making it difficult to realize a continuous oral hygiene evaluation system utilizing mouth-rinsed water closed in the oral cavity. As with mouth-rinsed water, saliva is a notable oral biomarker that has various health-related information [15]. Saliva also contains bacteria by retention in the oral cavity. This bacteria contamination mechanism is similar to that of mouth-rinsed water, and the high correlation between the bacteria in saliva and those in mouth-rinsed water [12] indicates that salivary turbidity can be a promising indicator of oral hygiene. Because saliva is naturally produced in daily life without any external operations, a closed measurement system, such as an intra-oral wearable sensor [16], can be realized in the oral cavity.

In this study, we proposed a mouthguard (MG)-type wearable sensor that utilizes salivary turbidity as an indicator for oral hygiene, which realizes continuous unconstrained monitoring of oral hygiene in the oral cavity. In the following sections, we investigated LED wavelengths and MG materials for optical measurement of salivary turbidity and constructed an MG-type salivary turbidity sensor. We also evaluated the turbidity quantification characteristics of the fabricated sensor and demonstrated the measurement of salivary turbidity with in vitro and in vivo experiments.

## 2. Materials and Methods

### 2.1. Fabrication of Mouthguard-Type Sensor

Figure 1a shows the configuration of the MG-type salivary turbidity sensor developed in this study. This sensor consists of a phototransistor (PT, APS3227SP1C-P22, Kingbright Electronic Co, Ltd., Taiwan, 3.2 mm × 2.7 mm × 1.1 mm, peak sensitive wavelength 580 nm), LED (SM1206UV-405-IL, Bivar Inc., Irvine, CA, USA, 3.2 mm × 1.4 mm × 1.6 mm), wireless module (customized, Discretek Inc., Shizuoka, Japan, 28 mm × 8.5 mm × 3.7 mm), and coin cells (SR716SW, Panasonic Corporation, Osaka, Japan) sealed inside a double-layered MG. The PT and LED were installed facing each other on the lingual surface of the incisor. This sensor evaluated the salivary turbidity by measuring the transmittance of light emitted from the LED through the saliva flowing in the gap between the LED and PT.

The MGs were vacuum-formed on plaster dental models obtained from subjects. First, the inner layer MG, which is in contact with the dentition, was formed using a dental vacuum-forming machine (Vacuum adapter I, Yamahachi Dental MFG., Co., Aichi, Japan). To achieve the desired shape of the turbidity-sensing part and to create a space capable of housing sensor elements such as PT and LED, a 3D-printed dummy part with the dimensions shown in the enclosed view of Figure 1a was fabricated. The outer layer MG was vacuum-formed on the plaster model with the dummy part, wireless module, and batteries were attached with double-sided tape (NW-N30, Nichiban Co., Ltd., Tokyo, Japan) at the designated position. Because the gap between the PT and LED in the turbidity-sensing part of the outer layer MG was incompletely followed in shape by only vacuum forming, the outer layer MG was locally reshaped by pressing a metal piece heated to about 100 °C into the gap. After forming the inner and outer MGs, each sensor element was connected using enameled wire with a diameter of 0.1 mm and sandwiched between the MGs at the appropriate position. Each sensor element was wired as shown in Figure 1b. Two coin cells for lighting the LED were connected in series with a 10 kΩ resistor and placed on the facial surface of the left molars. Finally, the edges of both MGs were heated with a heating gun (No. 880B, Hakko Corp., Osaka, Japan) and thermally welded to seal the sensor elements inside the double-layered MG. The fabricated MG-type sensor is shown in Figure 1c.

### 2.2. LED Wavelength

The wavelength of the LED utilized in the MG-type sensor was determined to be suitable for salivary turbidity measurement. The light transmittance of saliva at each wavelength was compared with that of formazine standard solution (FSS, Kishida Chemical Co., Ltd., Osaka, Japan), which is commonly used as a turbidity standard solution because of its superior particle uniformity and dispersibility [17]. Optical density (OD), representing logarithmic ratio of incident light intensity $I_{0}$ to transmitted light intensity I was utilized as an index of light transmittance, expressed as follows:(1)$$OD=-log(I/I_{0})$$

The OD spectrum of a saliva sample and FSS prepared to a turbidity of 553 FTU (formazine turbidity unit, where 1 FTU equals formazine at 1 mg/L) was measured using a microvolume spectrophotometer (Nanodrop™ 2000, Thermo Fisher Scientific, Waltham, MA, USA) with a resolution of 1 nm wavelength. The saliva samples were collected using a saliva collection aid (#5016.02, Salimetrics, LLC, Carlsbad, CA, USA). In addition, several FSS were prepared with the specified turbidity, and the light transmittance of FSS with 1–4000 FTU was evaluated especially at 660 nm, a wavelength widely used in industrial turbidimeters [18,19], and at 405 nm, a wavelength used in dental treatment [20].

### 2.3. Mouthguard Material

The light emitted from the LED passes through the outer MG twice before it is detected by the PT. To achieve high sensitivity, a reduction of light absorption at the MG is required, so the MG material used for the sensor was investigated.

The light transmittance of commercially available MG sheets of different materials was measured with a UV-visible spectrophotometer (V-530, Jasco Corp., Tokyo, Japan). As specimens, four kinds of MG materials were tested: polyethylene terephthalate-glycol (PETG, Erkodur J80421, ERKODENT Erich Kopp GmbH, Pfalzgrafenweiler, Germany), ethylene-vinyl acetate (EVA, Erkoflex J8041, ERKODENT Erich Kopp GmbH, Pfalzgrafenweiler, Germany), polyester (PEs, Ortholy 70730000, GC Ortholy, Tokyo, Japan), and polypropylene (PP, Essix^®^ type C 08100001, Dentsply Sirona K.K., Tokyo, Japan).

In addition, a three-point bending test was conducted using a tensile and compression tester (SV-55C, Imada Seisakusyo Co., Ltd., Aichi, Japan) to obtain stress–strain curves of MG materials.

### 2.4. Turbidity Quantification with Fabricated Mouthguard-Type Sensor

The fabricated MG-type sensor was immersed in FSS prepared at 1–4000 FTU, and the turbidity quantification characteristic of the sensor was evaluated from the output for each turbidity. Also, tap water was used as the 0 FTU sample.

The wireless module for the MG-type sensor functions as an A/D converter, potentiostat, and wireless transmitter using Bluetooth Low Energy (BLE), similar to those in the previous paper [21,22]. A constant voltage of 800 mV was applied to the PT using the wireless module, and the output current was evaluated as an output representing turbidity. In the potentiostat of the wire module, the voltage across a 10 kΩ shunt resistor connected in series to the PT is amplified 1.52 times using a non-inverting amplifier. The output current was measured in real-time after converting it to a digital signal with an A/D converter. The nominal resolution of the current measurement was 8 pA. As the definition in Section 2.2, OD was also used as an index of light transmittance. The intensity of the incident and transmitted light was represented by the output current for tap water (0 FTU) and the output current at each turbidity, respectively. The output current was measured by transmitting it via BLE wireless communication to a receiver connected to a PC. The sampling time of the wireless measurement was one second.

### 2.5. In Vitro and In Vivo Measurement of Salivary Turbidity

The OD value of saliva samples collected from subjects was evaluated by using a fabricated MG-type sensor and a microvolume spectrophotometer, respectively. For the MG-type sensor, 1 mL of saliva sample was dropped onto the sensitive part of the sensor, and the output current was measured wirelessly. The salivary turbidity was estimated from the calibration equation between OD and turbidity determined experimentally in advance. The validity of salivary turbidity measurement by the fabricated MG-type sensor was evaluated by comparing the estimated turbidity with that of the microvolume spectrometer.

After that, the MG-type sensor was attached to subjects, and in vivo turbidity measurement experiments were conducted. The subjects held tap water and saliva in their mouths, and the output current was measured, respectively. By utilizing the output current with tap water as a baseline, salivary turbidity was calculated using the calibration equation. The calculated results were compared with the results derived from the microvolume spectrometer with saliva samples collected after the in vivo experiment.

These experiments were conducted with the approval of the ethics review committee at Tokyo Medical and Dental University (TMDU) in accordance with the latest version of the Declaration of Helsinki (approval number D2018-054). Before the experiments, participants were provided with a detailed explanation of the methods and significance of the study, and written informed consent was obtained from participants. The study included a total of three participants in their 20s. Initially, participants were requested to complete a questionnaire regarding their gender, age, and body mass index.

## 3. Results and Discussion

### 3.1. Optical Wavelength for Salivary Turbidity Measurement

Figure 2a shows the OD spectrum results of the saliva sample and FSS prepared to 553 FTU, both measured using a microvolume spectrophotometer. Both the saliva sample and FSS exhibited higher OD values in the low-wavelength region. However, while the saliva sample demonstrated a distinct peak in OD around the wavelength of 290 nm, no such peak was observed in the case of FSS. This peak represents the absorbance characteristics of the proteins in saliva. Beyond the wavelength of 310 nm, the OD spectrum of the saliva sample and FSS were in good agreement. These results indicate that FSS can be used as an alternative sample to saliva for turbidity measurement by using light with a wavelength of 310 nm or longer. Subsequently, the turbidity dependence of OD with specific wavelengths was evaluated using FSS prepared at varying turbidity levels (1–4000 FTU). The results are shown in Figure 2b for the wavelength of 660 nm, which is used in industrial turbidimeters, and for the wavelength of 405 nm, which is used in dental treatment. Both wavelengths showed good linearity of OD with turbidity. As also indicated by the results in Figure 2a, the lower wavelength of 405 nm showed higher OD values at each turbidity level, which were 12 times higher than that of 660 nm. From these results, an LED with a wavelength of 405 nm, which is considered to be more sensitive for turbidity measurement, was selected as the light source for the MG-type sensor.

### 3.2. Mouthguard Material Appropriate for the Proposed Sensor

The light transmissivity of various MG materials was investigated to select the appropriate material for the proposed MG-type sensor. Figure 3a shows the light transmittance spectrum of each commercially available MG material. Each MG material exhibited a decline in transmittance at shorter wavelengths. Notably, at the selected wavelength of 405 nm for the sensor, the order of transmittance was as follows: EVA (79.9%), PETG (78.6%), PEs (77.6%), and PP (31.8%). To achieve the high-sensitivity turbidity measurement with the proposed MG-type sensor, the light emitted from the LED must pass the turbidity sensing part without being absorbed by the MG material. Therefore, PETG and EVA, which exhibited particularly high transmittance, were evaluated for their mechanical characteristics. Figure 3b shows the stress–strain characteristics of MG materials obtained from the three-point bending test. The stiffness of PETG and EVA were 0.58 N/mm and 0.079 N/mm, respectively, with PETG exhibiting 7.4 times stiffer than EVA.

According to these results, EVA was used for the inner layer MG, which is in contact with the gingiva and not involved in the optical path for turbidity sensing, because of its flexibility leading to low discomfort even in long-term wearing. On the other hand, for the outer layer MG, PETG was used because of its excellent light transmittance and high stiffness durable enough for dental occlusion.

### 3.3. Output Dependence on Turbidity

Figure 4 shows the output current of the fabricated MG-type sensor when immersed in FSS prepared at each turbidity level. The average output current was 12.8 μA when the sensor was immersed in tap water (0 FTU), serving as a baseline, and the output current decreased with increasing turbidity. The OD values for each turbidity level were calculated by substituting the obtained output current values into Equation (1). From the experimental results, the following calibration equation for turbidity and OD values was determined:(2)$$OD=4.0\times 10^{-4}\times {\left(turbidity\left[FTU\right]\right)}^{0.78}$$

The calibration equation elucidates the correlation between turbidity and OD across a wide turbidity range, covering the turbidity range of saliva (400–800 FTU). The coefficient of variation (C.V.) of the five measurements at 500 FTU was 6.4%. These results suggest that the proposed MG-type turbidity sensor has excellent reproducibility and turbidity calibration capability and is promising for salivary turbidity measurement.

### 3.4. In Vitro Turbidity Measurement of Saliva Sample

The turbidity measurement result when the saliva sample collected from a subject was dropped and held in the turbidity-sensing part of the MG-type sensor was 553 FTU, as shown in Figure 5. Also, substituting the OD value of the same saliva sample measured with the microvolume spectrophotometer into the following equation derived from the results indicated by the black line in Figure 2b, the turbidity of the saliva sample was calculated to be 576 FTU:(3)$$OD=2.78\times 10^{-4}\times (turbidity\left[FTU\right])+1.66\times 10^{-3}$$

The results were in close agreement with each other, indicating that the fabricated MG-type sensor can measure the turbidity of saliva samples.

### 3.5. In Vivo Turbidity Measurement of Saliva

The MG-type sensors were attached to subjects, and the output current was wirelessly measured when saliva and tap water were held in the mouth, respectively, as shown in Figure 6a. In this experiment, MG-type sensors were fabricated for each of the three subjects based on their dentition. The MGs were thin and fabricated to follow each dentition, contributing to less discomfort when it was worn. To eliminate the effect of fabrication errors, the calibration equations for each MG-type sensor were respectively identified. Figure 6b shows an example of the output current measurement results from the experiment. In this case, for the first 20 s shown in the figure, there was no liquid in the turbidity-sensing part; after it was filled with tap water or saliva, the output current changed. The output current was 11.9 μA for tap water and 10.8 μA for saliva, corresponding to 0 FTU and 374 FTU, respectively, when substituted into the identified calibration equation. The salivary turbidity measured with the MG-type sensors and microvolume spectrophotometer for the three subjects are shown in Table 1. The salivary turbidity measured with the MG-type sensors showed appropriate values with the reported salivary turbidity levels and measurement results with the spectrophotometer. The average salivary turbidity among the three subjects was 457 FTU.

## 4. Conclusions

In this study, we developed the MG-type salivary turbidity sensor to realize a continuous and unconstrained measurement of salivary turbidity, which is an indicator of oral hygiene. To realize the high-sensitivity optical measurement of salivary turbidity with the MG-type sensor, the wavelength of the light source and MG material were experimentally optimized. The turbidity quantification characteristic of the MG-type sensor was evaluated using FSS, and a dynamic range of 1–4000 FTU was achieved, which covers the typical salivary turbidity levels.

In vitro and in vivo turbidity measurement results of saliva samples showed similar results with the MG-type sensor as with optical measurements with a spectrophotometer, indicating that the MG-type sensor is capable of measuring salivary turbidity. In addition, the in vivo measurement experiments demonstrated wireless unconstrained with the fabricated MG-type sensor.

The proposed MG-type sensor is a wearable device that can measure salivary turbidity in the oral cavity, contributing to the realization of real-time continuous evaluation of oral hygiene. In future work, we will evaluate the feasibility of long-term continuous salivary turbidity measurement with the MG-type sensor, and the responsibility of the MG-type sensor for real-time changes in turbidity.

## Figures and Tables

**Figure 1 sensors-24-01436-f001:**
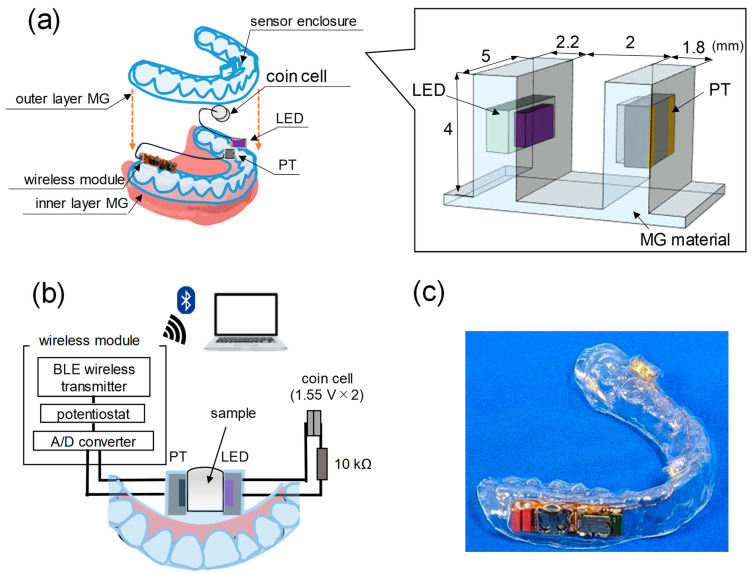
(**a**) Schematics of configuration of proposed MG-type salivary turbidity sensor and enlarged view of turbidity-sensing part; (**b**) circuit diagram and data transfer system of MG-type sensor; (**c**) photograph of fabricated MG-type turbidity sensor.

**Figure 2 sensors-24-01436-f002:**
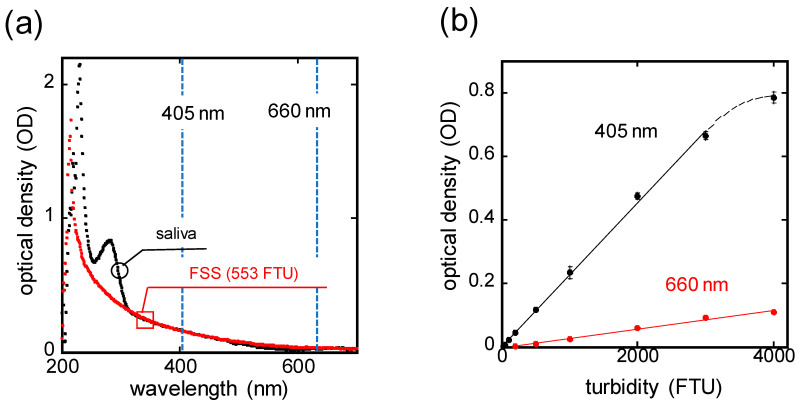
(**a**) OD spectrum of a saliva sample and FSS prepared to 553 FTU; (**b**) turbidity-dependent characteristics of OD for wavelengths of 405 and 660 nm on FSS.

**Figure 3 sensors-24-01436-f003:**
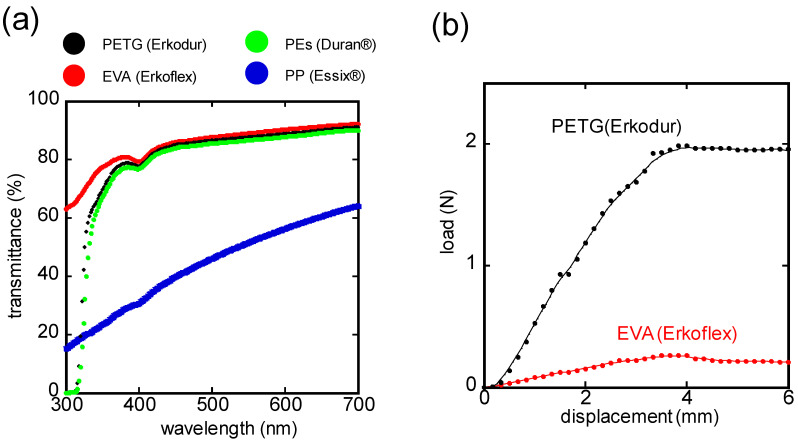
(**a**) Transmittance spectrum of some commercial MG materials; (**b**) mechanical characteristics of MG materials (PETG and EVA) obtained from three-point bending test.

**Figure 4 sensors-24-01436-f004:**
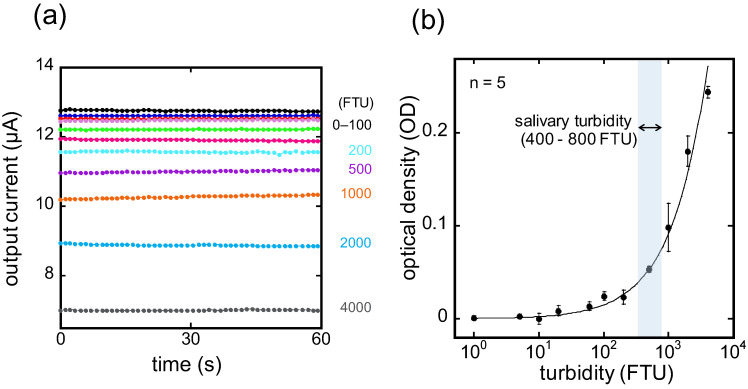
(**a**) Output currents of MG-type sensor to FSS for various turbidity levels; (**b**) identified correspondence between output OD of the MG-type sensor and turbidity level.

**Figure 5 sensors-24-01436-f005:**
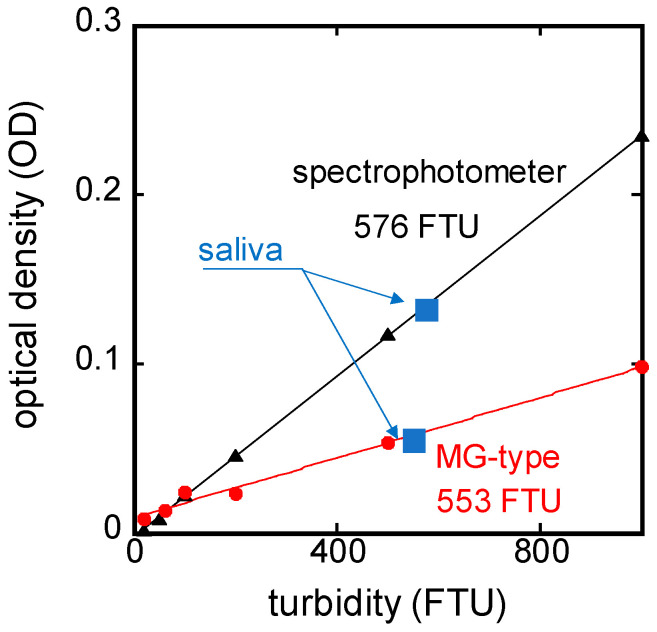
Result of in vitro measurements of saliva sample with microvolume spectrophotometer and MG-type sensor.

**Figure 6 sensors-24-01436-f006:**
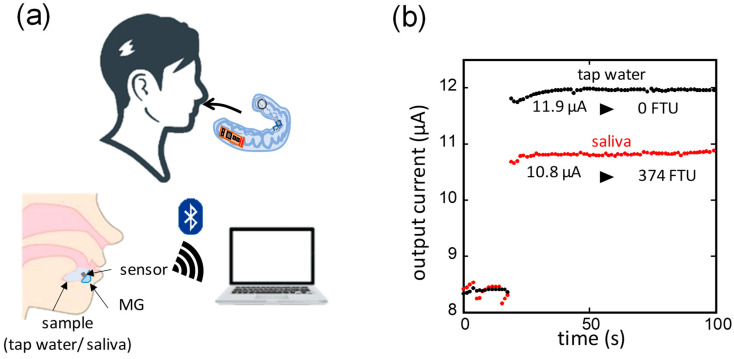
(**a**) Schematics of in vivo experiment of wireless turbidity measurement; (**b**) output current of MG-type sensor for tap water and saliva in the oral cavity.

**Table 1 sensors-24-01436-t001:** Salivary turbidity levels measured with MG-type sensor and spectrophotometer.

Subject	MG-Type Sensor (FTU)	Spectrophotometer (FTU)
# 1	374	435
# 2	414	430
# 3	583	620

## Data Availability

The data supporting this study’s findings are available from the corresponding author (K.M.) upon reasonable request.

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
