# Peer review of "Mouthguard-Type Wearable Sensor for Monitoring Salivary Turbidity to Assess Oral Hygiene"

_sensors, 2024, doi:10.3390/s24051436_

Round 1
Reviewer 1 Report
Comments and Suggestions for Authors
This manuscript studies a mouthguard-type wearable sensor designed for monitoring salivary turbidity. The sensor employs LED and a photodetector to detect turbidity by measuring changes in current. While the concept is intriguing, the novelty is not clearly defined. Several points warrant clarification:
1. The manuscript mentions a distinct peak near the wavelength of 290 nm. Elaboration on the significance of this peak is essential for a comprehensive understanding.
2. The technical details of calculating the real-time average current in the measurement scheme need further elucidation. Additionally, clarity on the stability and resolution of the measurements is crucial.
3. The method for varying turbidity is not clearly explained. Are multiple liquids with verified turbidity levels used for this purpose?
4. As mentioned in Line 241, it is imperative to conduct an in-depth exploration of how variations in dentition shape can affect measurement outcomes.
5. Based on the turbidity measurements in Table 1 across different subjects, is there a correlation between turbidity readings and the health status of the subjects?
6. The meaning and relevance of the curve depicted in the bottom left corner of Figure 6b require clarification.
7. The manuscript mentions the use of a spectrophotometer for spectrum measurement, but a scheme outlining the measurement process is conspicuously absent from the figures.
Addressing these points would enhance the clarity and depth of understanding regarding the mouthguard-type wearable sensor and its application in salivary turbidity monitoring.
Author Response
We appreciate you taking the time to review our article. We modified the manuscript according to all reviewer's comments and suggestions. Please see the attachment. The modified parts are indicated with yellow marker.

Reviewer 2 Report
Comments and Suggestions for Authors
The manuscript is well prepared and clearly written. The results are very important for practical applications.
Author Response

(The authors gave the same response as above.)

Reviewer 3 Report
Comments and Suggestions for Authors
The authors proposed the wearable mouthguard-type sensor that evaluates salivary turbidity by measuring the light transmittance of salvia with an LED and a phototransistor. I think the research topic is interesting, and the experimental flow is well constructed. To get the manuscript in better shape, I have several concerns listed below.
1. In general, a LED source has the Lambertian radiation distribution. Can you explain how the radiation pattern of LED changes in air and in salvia? Also, can you explain how changes in LED radiation distribution affect the OD value?
Author Response

(The authors gave the same response as above.)

Reviewer 4 Report
Comments and Suggestions for Authors
In this study, authors developed the MG-type salivary turbidity sensor to realize a continuous and unconstrained measurement of salivary turbidity, which is an indicator of oral hygiene. To realize the high-sensitivity optical measurement of salivary turbidity with the MG-type sensor, the wavelength of light source and MG material were experimentally optimized. Overall, the paper is well written, the experimental studies are well designed and the results and analysis are sound. The manuscript can be published after addressing the following comments:
1. Line 35, in the part of ‘Keywords’, the selection of keywords is inaccurate, recommending to add Oral Hygiene, and replace ‘turbidity; saliva’ with ‘salivary turbidity’;
2. Line 85, Please add a space between "Figure.1" and "(a)";
3. Line 93, Please add a space between "Figure.1" and "(b)";
4. Line 214, change ‘Eq. (1)’ to ‘Eq. (2)’ ???
5. Line 255, please change ‘Conclusion’ to ‘Conclusions’;
6. Will the mouthguard-type wearable sensor cause discomfort to the human body while using for monitoring, if not, please clarify.
7. The number of references is small, please add some references.
Author Response

(The authors gave the same response as above.)

Round 2
Reviewer 1 Report
Comments and Suggestions for Authors
Thank you for the responses, and I don't have any additional comments or suggestions.